# The Current State of Deep Brain Stimulation for Chronic Pain and Its Context in Other Forms of Neuromodulation

**DOI:** 10.3390/brainsci8080158

**Published:** 2018-08-20

**Authors:** Sarah Marie Farrell, Alexander Green, Tipu Aziz

**Affiliations:** 1Nuffield Department of Surgical Sciences, John Radcliffe Hospital, University of Oxford, Oxford OX3 9DU, UK; sarah.farrell@medsci.ox.ac.uk (S.M.F.); alex.green@nds.ox.ac.uk (A.G.); 2Nuffield Department of Clinical Neurosciences, John Radcliffe Hospital, University of Oxford, Oxford OX3 9DU, UK

**Keywords:** pain, DBS, ACC

## Abstract

Chronic intractable pain is debilitating for those touched, affecting 5% of the population. Deep brain stimulation (DBS) has fallen out of favour as the centrally implantable neurostimulation of choice for chronic pain since the 1970–1980s, with some neurosurgeons favouring motor cortex stimulation as the ‘last chance saloon’. This article reviews the available data and professional opinion of the current state of DBS as a treatment for chronic pain, placing it in the context of other neuromodulation therapies. We suggest DBS, with its newer target, namely anterior cingulate cortex (ACC), should not be blacklisted on the basis of a lack of good quality study data, which often fails to capture the merits of the treatment.

## 1. Introduction

Chronic pain is an important health issue drastically altering the lives of those it affects; it is estimated that 5% of the population suffer chronic pain despite pharmacotherapy [1]. The ramifications include the mental health status of the individual in terms of emotional well-being [2], opioid dependency [3], and cognitive function [4]. The socioeconomic sequelae include loss of productive workforce [5]. In the United States it is estimated to cost $500 billion a year in medical treatment and loss of productivity, with an estimated 116 million people suffering from the condition [6].

Neuropathic pain is defined as “pain caused by a lesion or disease of the somatosensory system” [7]. Chronic pain is that extending beyond the time of injury and healing. Much has been made of the types of pain amenable to different neuromodulation methods. It has been tempting to categorize chronic pain by its cause, and then into categories such as ‘nociceptive vs. deafferentation’ or ‘peripheral vs. central’. The utility of this is questionable, given that the development and maintenance of pain is now thought to involve neuronal plasticity encompassing centrally mediated changes, as suggested by both functional neuroimaging and electrophysiological data [8,9,10,11,12,13]. The efficacious results of deep brain stimulation (DBS) in spinal cord-related patients, for example, those with failed back surgery syndrome (FBSS), are consistent with this theory, suggesting a centrally mediated component to this initially peripheral injury, which is able to respond favourably to thalamic or anterior cingulate cortex (ACC) stimulation [14]. The lesson here may be to worry less about the specific pain aetiology or the categorization of physical pain, and instead select patients whose pain is not complicated by psychogenic factors such as catastrophization or other negative predictors of good outcome [15].

## 2. Management of Chronic Pain

### 2.1. Pharmacotherapy and Non-Invasive Neuromodulation

Pharmacotherapy, such as opioids, carbamazepine, gabapentin, tricyclic antidepressants, and serotonin- and norepinephrine-selective reuptake inhibitors, often fail those afflicted [16]. They cause side-effects such as sedation and nausea because of the non-specificity of the medication, and opioids in particular suffer from reduced long-term efficacy due to receptor downregulation. Others may focus on reducing aberrant neuronal activity in peripheral nerves, failing to address the central nervous system (CNS) aspect involved in its development and maintenance. Neurosurgical attempts to relieve chronic pain focus on the various structures in the pain pathway (peripheral nerve, dorsal root, spinal cord, midbrain, thalamus, and cingulate cortex), either lesioning, electrically stimulating, or perfusing with analgesia/anaesthetic. The opiate epidemic in the USA has forged a renewed interest in neuromodulation. Between 2000–2012, the prescription use of opioids tripled [17]. In 2016, 42,000 Americans lost their lives to opioid overdose, with fentanyl the biggest culprit [18]. Whilst heroin takes second place, it is thought that the indiscriminate prescription of opioids encourages those predisposed to develop addictions, which leads to more recreational drug use [17,18,19].

Public consciousness is more at ease with less invasive approaches. Non-invasive neuromodulatory strategies include repetitive transcranial magnetic stimulation (rTMS) and transcranial direct current stimulation (tDCS). Both are thought to alter the maladaptive plasticity within pain circuits, affecting the nuclei in the thalamus and subthalamic regions [20,21,22]. The effects of rTMS are thought to be modest and short-lasting [23]. A systematic review of six studies assessing 127 patients treated with rTMS following spinal cord injury (SCI) concluded that, despite some reduction in the pain indices following rTMS, the effects were unable to reach statistical significance [24]. Of course, the field of non-invasive modulation has its own discrepancies that may cloud the literature; the location (motor cortex versus premotor cortex/dorsolateral prefrontal cortex), type, and orientation of the coil, schedule of repetitive stimulation, and persistence of therapeutic response. rTMS has also been shown to predict the beneficial effects of a more invasive longer-term treatment [25,26], namely motor cortex stimulation (MCS), discussed below, allowing for an exciting area of future research regarding the pre-selection of patients. The second non-invasive approach, tDCS, in contrast to rTMS, does not result in neuronal firing, but changes the resting membrane potential, thereby altering the neuronal excitability. It is thought to alter neurotransmitter systems, hence its longer-term potential [27]. A positron emission tomography (PET) study of 16 SCI pain patients who were administered stimulation over the motor cortex demonstrated not only a reduction in the pain visual analogue scale (VAS) scores, but were found to exhibit an altered metabolism in the subgenual ACC, left dorsolateral prefrontal cortex, and insula, suggesting an effect of tDCS on the affective component of pain processing [28]. As a result of these studies, tDCS is listed as a third-line therapy for neuropathic pain for SCI in the CanPain guidelines [29]; the only neuromodulatory strategy to be included.

### 2.2. Spinal Cord Stimulation

Spinal cord stimulation (SCS) is a successful and common strategy for treating chronic pain, first reported half a century ago [30,31]. Classically, it is thought to activate the large rapidly conducting Aβ fibres; leading to the potentiation of inhibitory neurons on pain, as per Wall and Melzack’s theory [32]. The efficacy of alternative stimulation waveforms such as high frequency and burst, however, suggest that a revision of this theory is needed [33]. For conventional SCS, electrodes are placed with the stimulating tips between the C5 and T1 vertebral bodies for upper limb pain, and between T9 and T11 for lower extremity pain. Traditional ‘tonic’ stimulation induces a paraesthesia covering the anatomical distribution of the pain, although newer waveforms are paraesthesia-free. SCS is excellent for aetiologies such as FBSS, multiple sclerosis, and complex regional pain syndrome (CRPS), but less effective for phantom limb pain and postherpetic neuralgia [34]. The large number of patients, low morbidities, and improvements in technology over the last 50 years have led to its widespread use. It is the most demonstrably successful neurostimulation method used for chronic pain, largely due to the upsurge of patients with FBSS, present in 10–40% of patients after lumbar spine surgery [35,36].

The first randomised control trial (RCT) looking at SCS for chronic pain used FBSS patients, with the control group receiving repeat lumbar spine surgery. North et al. found a significantly greater number of patients with a 50% or greater pain relief (9/19 for SCS, 3/26 for controls; *p* < 0.01) [37]. The positive results were followed by the PROCESS trial; an RCT of 100 patients; controls received conservative medical management (CMM). SCS delivered better pain outcomes at 6, 12, and 24 months follow up, with the percentage of patients reaching the target of 50% reduction at 24 months being significantly higher; 37% vs. 2% (*p* = 0.003) [38,39]. Indeed, the preliminary findings of a further RCT (*n* = 218) comparing optimal medical management and SCS showed SCS to be superior in the number of patients to reach a 50% pain reduction [40].

Success with other pain aetiologies has been published, but is less convincing. Kemler et al. demonstrated that patients with complex regional pain syndrome (CRPS) experienced a mean VAS reduction of 2.4/10 cm (*n* = 24) at a six month follow up, and 3.6/10 cm (*n* = 18) for those with physical therapy plus SCS compared to an increase in 0.2 cm/10 cm at six months for physical therapy alone [41]. These values, however, do not include the 12 patients from the SCS arm who failed to complete the implant procedure because of an unsuccessful test stimulation. At a five-year follow-up, the pain relief differences did not reach significance. [42]. Two prospective studies suggest that SCS improves painful peripheral neuropathy compared to medical management with 60% of patients in the SCS group reaching the success criteria at six months, compared to 5% and 7% in the control arm [43,44].

The level of efficacy for SCS in chronic pain is still classed as ‘moderate’, but importantly, it is shown to be safe with a 2005 systematic review of studies showing no major adverse events [45]. Novel devices are demonstrating further potential. High frequency versions of the treatment provide up to 10,000 Hz (compared to up to 1200 Hz) and have demonstrated positive results in feasibility studies [46]. These versions have even been shown to be more successful compared to the conventional frequencies in a study of 193 subjects (171 of whom completed implantation) of back pain (84.5% at three months with high frequency compared to 43.8% with conventional) and leg pain (83.1% vs. 55.5%; *p* < 0.001) [47,48]. Burst DR^®^ stimulation (Abbott, One St Jude Medical Drive, St Paul, MN 55117, USA) has also enjoyed additional success with hints it might improve on conventional SCS outcomes with similar safety profiles in both FBSS [49] and CRPS [50]. Dorsal root ganglion (DRG) stimulation has some theoretical advantages over conventional SCS; it delivers stimulation directly to the nerve roots and is less vulnerable to positional changes [51], as such initial studies have shown positive results in the regions not usually successful with SCS, for example CRPS and groin pain. The prospective RCT that led to United States Food and Drug Administration approval of its use demonstrated that at a three month follow up 81.2% of CRPS/causalgia patients treated with DRG achieved success (defined by greater than 50% reduction in VAS score) with conventional stimulation success at 51.7% [52]. However, higher procedural adverse events were found in the DRG group, such as pain at the incision site (7.9% DRG and 6.9% SCS).

Patient selection is key, as for all of the neurostimulation procedures, with risk factors for negative results, including opiate addiction, catastrophization, active depression/anxiety, low-activity levels, as well as ongoing litigation, necessitating a role for presurgical neuropsychological evaluation [53]. Currently, rTMS promises insight into finding the suitable candidates [54]. If SCS fails, or if the pain aetiology is central (e.g., post-stroke pain and atypical facial pain), a surgeon may try either DBS or MCS, largely depending on their skillset and familiarity, as has been shown to be the case in several studies [55,56,57].

### 2.3. Deep Brain Stimulation

Deep brain stimulation provides a further opportunity to alleviate pain in some individuals. Specific indications include central post-stroke pain, atypical facial pain, brachial plexus injury, and some patients who have failed SCS. Such conditions do not generally respond to SCS or DRG stimulation, except for some cases of facial pain that may respond to peripheral nerve stimulation or high cervical stimulation [58,59]. There are three main DBS target sites, namely: (1) the thalamus- ventral posterolateral nucleus and ventral posteromedial nucleus (VPL/VPM); (2) regions surrounding the third ventricle and aqueduct of Sylvius, including the grey matter- Periventricular grey and periaqueductal grey (PVG/PAG); and (3) the newer target of the rostral anterior cingulate cortex (ACC) posterior to the anterior horns of the lateral ventricles. A fourth target, the posterior hypothalamus, may be considered specifically for cluster headache (not discussed in detail in this article), in those for whom occipital nerve stimulation has failed. For the thalamus and PAG, the DBS at lower frequencies (<50 Hz) is thought to be analgesic and at higher frequencies (>70 Hz) is thought to be hyperalgaesic. When VPL/VPM is targeted, pleasant paraesthesia supplants a painful sensation, whereas PVG/PAG stimulation induces a sense of warmth and analgesia over the area of pain [60,61,62]. ACC stimulation is thought to remove the affective aspect of pain, and thus high frequencies have shown to be clinically effective [63]. The exact mechanism of action is equivocal [64]. There are mixed reviews regarding endorphin and opioid pathway theories, with earlier studies showing these to be less likely [65,66]. However, more recent positron emission tomography (PET) studies demonstrate a reduced binding of (^11^C)diprenorphine (a ligand with high opioid affinity) in the dorsolateral PAG when DBS electrodes were switched on vs. off, indicating the DBS-induced PAG release of endogenous opioid peptides [67,68,69]. However, Pereira et al. have shown elevated gamma band frequency in PAG/PVG upon the administration of naloxone in DBS patients [70], suggesting an enhanced awareness of the patient’s worsening pain. Other studies suggest that both DBS and SCS may modulate the gene expression [71,72].

A brief history of the evolution of DBS for chronic pain helps to contextualise how such an invasive procedure becomes relevant to the field of chronic pain. The ability of DBS to alleviate pain was first seen in septal self-stimulation experiments in rodents [73]. The first DBS was performed for nociceptive pain in the 1950s [74]. The 1960s saw use of DBS to alleviate pain in cancer patients [75]. Further impetus for electrical stimulation, initially in the form of peripheral nerve stimulation [76] and then spinal cord [30] stimulation, was found in Melzack and Wall’s gate theory of pain [32]. By the 1970s, several centres were performing DBS for neuropathic pain. Evidence for targeting PVG/PAG came from pain relief in rodents during awake surgery [77,78], and human studies followed [79,80,81,82]. Evidence for targeting the thalamic nuclei (VPL/VPM) came from ablative surgery [83,84,85], with subsequent human studies [86,87,88,89,90], along with Adams, who also targeted the internal capsule [91,92,93], moving to more medial targets developed from localisation errors and investigations in the current spread [94,95,96,97]. By 1987, out of the 141 patients implanted, 59% obtained initial pain relief, although this percentage reduced to 31% at follow up (mean 80 months) [98]. The major complications were listed as one death, 12% wound infections, and 3.5% intracranial haemorrhage.

The initial human studies lacked numbers and were marred by the variability of the surgical technique or settings, different locations used, and different pain profiles of the patients being treated, leading to heterogenous patient groups, and hence the studies were not successful. An early RCT by Marchland showed placebo improved pain intensity, yet the stimulation of the thalamus did not [99]. Two failed industry open label studies further dampened the excitement. The first (*n* = 196), by Medtronic, was powered to show if half of the patients that were internalised would get at least 50% pain relief. This failed to reach the outcome. The second (*n* = 50) trial failed because of the lack of accrual [100]. Consequently, the FDA afforded only ‘off-label’ status to DBS for chronic pain relief [101]. Few clinical trials have been reported since, despite the apparent need to rectify the multitude of issues plaguing the above trials. These issues include the nonrandomised nature, heterogenous case mix, subjective and unblinded assessment of patient outcome, inconsistencies in DBS sites stimulated, stimulation parameters chosen, and number of electrodes implanted.

Despite this relegation to the ‘off-label’ status, the DBS for chronic pain has yielded many success stories, arguably not all represented in the literature. Boccard et al. reported the long-term outcome of 59 patients with a variety of pain aetiologies receiving DBS in the sensory thalamus, periventricular grey, or both. After a mean follow-up of almost 20 months, the pain was compared to the pre-operative levels. Improvement was defined as a global improvement of their EuroQol-5D (EQ-5D). For patients with phantom limb, 8/9 improved; for brachial plexus injury, 3/6 improved; for post-stroke pain, 16/23 improved; for spinal cord injury, 4/7 improved; and for cephalalgia, 6/11 improved [102]. For the patients that improved, the pain was reduced by 50% on a visual analogue scale.

In another study of mixed pain aetiology, Kumar et al. reported the outcome of DBS in the periventricular/periaqueductal grey area (*n* = 49) or sensory thalamus/internal capsule (*n* = 16) [103]. Mean follow-up was 78 months and success defined as a greater than 50% reduction in VAS pain scores. For the patients with FBSS, 32/43 had long-term improvement; for peripheral neuropathy, 3/5 improved; for thalamic pain, 1/5 improved; for trigeminal neuropathy, 4/4 improved; for spinal cord injury, 0/3; for post-herpetic neuralgia, 0/3; and for phantom limb pain, 1/1 improved.

A 2005 meta-analysis listed differing success rates depending on aetiology. Most of the studies included defined pain in the normative way, of “at least 50% of patients with a 50% improvement in pain scores”. Some pain aetiologies fared better than others. For example, FBSS has a 78% success rate (pre-internalisation), causalgia 80%, cancer 65.2%, lumbosacral radiculopathy 90.5%, and lumbar arachnoiditis 77.8%; however thalamic central lesion, phantom limb, and cervical root/brachial plexus lesion scored less well (31.1%, 44.4%, and 50%, respectively). The authors calculated the total success rate overall to be 232/424 (54%) of the surgeries, of which the successful ones were internalised. In the post-internalisation, 76.1% remained successful [61]. Rasche et al. further demonstrate the importance of careful patient selection, agreeing that DBS appears to be particularly useful for FBSS, although it shows disappointing results for SCI and poststroke pain [104].

In the past decade, to our knowledge, there have been only three clinical trials pertaining to DBS and pain. Two are prospective randomised crossover trials and one [105] is a non-randomized open label trial that serves as a comparison between MCS and DBS (discussed below). The more tangential of the RCTs targeted the subthalamic nucleus in 19 Parkinson disease patients to show that the post-operative pain threshold increased with no correlation between the increased pain threshold and improvements in the UPDRS-3 scores, thus suggesting that clinical pain alleviation after subthalamic nucleus-DBS is not just a by-product of the improvement in motor complications in these Parkinson patients [106].

The third RCT in a 2017 trial using post-stroke pain patients targeted the ventral striatum/anterior limb of the internal capsule. Ten patients, nine of whom progressed to internalisation, were implanted, waited one month, and then randomised to either ‘active’ or ‘sham’ stimulation for three months, after which they crossed over to the other treatment category. The results show no significant difference in pain-related-disability as indexed by the arbitrarily set ‘greater than 50%’ improvement on the pain disability index (11% DBS on vs. 12% DBS off; odds ratio = 1.05, 95% CI 0.96–1.15 *p* = 0.270). However, the authors highlight an acceptable safety profile, with 14 serious adverse events recorded and resolved, only three of which (one seizure, one wound dehiscence, and one wound infection) were identified as being related to study. They also found statistically significant improvements on multiple outcome measures related to the affective sphere of pain, for example, a 50% improvement on MADRAS (Montgomery–Asberg Depression Rating Scale); with 44% reaching the 50% reduction target with DBS on vs. 19% with DBS off; odds ratio 0.30, 95% CI 0.11–0.83; *p* = 0.020. No significant difference was seen for the VAS reductions between the on and off states [107].

These modest results may reflect the inability of the data to represent the potential for DBS for a number of reasons. DBS tends to be a treatment that takes place once SCS has failed, suggesting that the patient population that received DBS are filtered to be those that are more difficult to treat than those received by SCS, skewing the results unfavourably against DBS. It is also pertinent that for some of those patients who do not meet the arbitrary ‘50% reduction in pain’ threshold, testimonials suggest that even a partial reduction in pain has resulted in a greatly increased quality of life. Put simply, there are patients in whom DBS has reduced their pain and improved their subjective quality of life, but who are represented as a ‘fail’ in the literature. Reductions in VAS are poorly correlated with patient satisfaction or disability [108]. In fact, in this study, 5/9 patients said they would have the surgery all over again if they knew the result they would get—suggesting over 50% success rate according to patient satisfaction. There are further issues with the outcome measures used to detect successful results; the removal of a particular component of pain, for example, burning hyperesthesia, may unmask another type, such as muscular allodynia, as has been described after stroke [109]. In the future, a score capturing a more objective measure of the changes in analgesia may be useful. Investigations into the heart rate variability changes and blood pressure monitoring may provide an objective measure of efficacy that correlates to analgesia [110,111].

Furthermore, the levels of success rates would be easily increased following improvements in patient selection, in predicting who will respond. Diffusion tensor imaging (DTI) techniques (i.e., looking at network connectivity to predict a response) may provide a solution, as has been suggested from work with movement disorder patients. In these patients, it is thought that predominant beta-activity may come to serve as an electrophysiologically determined target for the optimal outcome in the subthalamic nucleus for Parkinson’s disease [112].

Fundamentally, RCTs examine population statistics—they look at mean changes. For an individual with refractory pain who has used up the limited treatment options available, it is a question of risk vs. benefit, for example, an individual may be weighing up a 20% chance of success with a 1:500 risk of stroke. It is reasonable the individual may choose to take this risk.

## 3. Anterior Cingulate Cortex: A More Recent DBS Target for Chronic Pain

Pain relief by cingulotomy [113,114,115] ignited interest in the dorsal ACC as a potential DBS target in the treatment of chronic refractory pain, especially for those with a substantial affective component to pain. Foltz and White built on the observations of psychiatric lobotomy patients to suggest that the transection of the cingulum bundle might benefit those patients with a disproportionally large affective component to their pain [116]. In the discussion of their findings, they reported a universal decrease in the distress associated with chronic pain. Ballantine and Hurt later introduced a stereotactic approach in 1966 [117,118]. This target is supported by more recent imaging and neurophysiological evidence describing the role of the anterior cingulate in the perception of pain [119]. Studies using functional Magnetic Resonance Imaging (fMRI) have demonstrated an increased activation of the dorsal anterior cingulate cortex (dACC) during both empathic and experienced pain [120], supporting the notion that the dACC is implicated in the affective component of pain. Furthermore, PET studies of thalamic DBS patients (*n* = 5) demonstrate the activation of anterior ACC throughout the 40 min of DBS tested, and a more posterior ACC activation at a delay (approximately 30 min) after onset [121].

In 2007, Spooner and colleagues, reported the first case in which standard DBS electrodes were used to administer high-frequency electrical stimulation of the dACC. The patient had refractory neuropathic pain resulting from a complete C4 level spinal cord injury, despite numerous medical and surgical interventions. Targets were place in the cingulate cortex (bilaterally) and PVG (unilateral), and a one-week blinded stimulation period was completed. The outcome measures included VAS, pain medication usage, and self-described mood. Both the PVG and ACC stimulation decreased the VAS pain rating and led to a reduction in the subcutaneous lidocaine usage [122]. At the three-month follow up, the ACC stimulation yielded a VAS score of three (out of 10) and a mood described as ‘best’. In comparison, PVG scored a value of four with an ‘average’ mood, and no stimulation resulted in a VAS score of 10 and a mood self-described as ‘worst’. The improvement in mood and reduced pain with bilateral cingulate stimulation implies that the affective component of pain is targeted. A series of cases from the Oxford group have since replicated these benefits in case series ranging from n-of-1 to n-of-24, demonstrating that bilateral ACC stimulation is not only efficacious for a variety of pain aetiologies including FBSS, post-stroke, spinal cord injury, brachial plexus lesions, and head injury, but that it delivers long term control over a period of years [63,123,124]. In the most recent case-series, 83% of patients showed an improvement in numerical rating scale (NRS) pain score by at least 60% (*p* < 0.001) at the six month follow up, and the McGill pain questionnaire (MPQ) decreased by 47% (*p* < 0.01). After one year, the NRS score decreased by 43% (*p* < 0.01), the EQ-5D quality of life measure was significantly reduced (mean, −30.8; *p* = 0.05), and significant improvements were also observed for different domains of the short form health survey (SF-36). At the longer follow-up, the efficacy was sustained up to 42 months in some patients, with an NRS score as low as three [63]. Importantly, the patients described that although pain was present, it was ‘less bothersome’ or ‘separate from them’, playing into the affective role of the ACC. This suggests that the NRS scores are not necessarily capturing the patient satisfaction of the treatment. Of note, four patients experienced problems with seizures/epilepsy after long-term stimulation, one of whom suffered from breakthrough seizures despite being off-stimulation and taking anti-epileptics. The ethical dilemma this poses has been discussed in a recent publication [125]. If this risk can be minimized, it is possible that ACC may be able to salvage patients in whom other neuromodulation has failed. However, given the small number of patients and short follow up time compared to other DBS targets, we must be cautious regarding the projections for its future use. The case studies and series pertaining to DBS targeting ACC for chronic pain are summarised in Table 1.

## 4. Motor Cortex Stimulation and Its Comparison with Deep Brain Stimulation

There has been much discourse surrounding the need to compare the less invasive motor cortex stimulation (MCS) with DBS in terms of efficacy and the side-effect profile. This discourse is yet to filter down to good quality studies of clinical relevance.

To briefly describe MCS, epidural electrodes are implanted over the motor cortex through a frontoparietal craniotomy. One or two electrodes are implanted, either parallel or orthogonal to the central sulcus, to comply with the motor representation of the painful area. Then, similar to DBS, the electrode is connected to a subcutaneous implant pulse generator, with stimulation parameters adjusted post-operatively.

The MCS studies show varying levels of success. In a summary of the literature surrounding MCS for chronic pain up to 2006, a greater than 40% improvement in pain scores were reported in 54% of 117 patients with central pain, and 68% of 44 patients with trigeminal neuropathic pain [101]. A prospective audit of 10 patients with mixed pain aetiologies showed a 50% success rate (relief of pain between 50–90% from baseline), with no clear predictability based on the mixed pain aetiologies [126]. Later results have shown mixed success. Two studies have effectively reported MCS to be ineffective, but are hampered by issues surrounded by patient retention, patient selection, and administration of the treatment [127,128]. Lefaucher et al. reported an RCT of MCS for peripheral neuropathic pain, where 13 patients had a significant reduction in some measurements of pain when the device was ‘on’ compared to ‘off’. However, these results were statistically insignificant after multiple comparison correction [129]. Nguyen et al. reported a randomized, blinded crossover trial of MCS in 10 patients with neuropathic pain with significant reduction in pain when the device was switched ‘on’ compared to ‘off’ [130]. Notably, a disappointing response was seen in hemibody post-stroke pain and post-herpetic neuralgia patients. The poor results for the post-herpetic neuralgia have been replicated [131]. The same group reported better results for patients with complex regional pain syndrome, however, with four out of five patients experiencing improvement with pain, sensory, and sympathetic symptoms [132]. Results concerning trigeminal neuropathic pain also appear more successful, with Rasche et al. demonstrating that 5 out of 10 patients received a reduction in VAS pain scores of at least 50 [133]. By 2012, a review of the MCS facial pain literature showed that an impressive 84% of 100 patients implanted following a trial had at least 40% pain improvement [134]. This success continues to be replicated, with 72% of 36 patients receiving MCS for trigeminal neuropathic pain showing a mean VAS reduction from 8.11 to 4.5 cm (*p* < 0.05) and a mean VAS score of 5 cm at the last follow up (mean 5.6 years and 26 patient included) [135].

Table 2 lists the studies, to our knowledge, involving a MCS vs. DBS comparison. As shown, the studies are sparse, and with a superficial glance, may seem to favour MCS—this, however, is far from clear cut. The study by Son et al. appears to favour MCS. Nine chronic pain patients of varying aetiologies were implanted with both MCS and thalamic DBS; 6/8 responded better to MCS and 2/8 responded to DBS [105]. They concluded MCS to be a reasonable initial means of treatment given the less invasive nature and the lack of evidence showing DBS to be of higher efficacy. This conclusion is perhaps premature, as the majority of DBS for chronic pain no longer involves solely a thalamic implant. It would be prudent to conduct similar studies with DBS implants in the PAG in addition to thalamus, or alternatively, the ACC if appropriate, in order to avoid a ‘MCS gold standard’ vs. ‘DBS old standard’ comparison. Ideally this type of RCT study would be replicated for a wider variety of pain aetiologies, specifically using DBS targets for which there is more experience, such as thalamus/PAG/ACC. In a review of the MCS and DBS literature, Honey et al. suggest that, in addition to having ‘pure cohorts’ comparing DBS and MCS in one condition at a time, future trials should incorporate a post-operative phase, where the success of each modulation type can be maximized, followed by a blind crossover phase to test response, and finally, an open-label phase to monitor long term efficacy [136].

For any given study, it is reasonable to suppose that the research group and/or surgical team in question is better skilled at one type of intervention over the other, producing unintentionally biased data. This may go some way to explaining why, even when controlling for pain type (i.e., just focussing on post-stroke pain patients), Katayama found MCS to yield higher success rates than DBS [56], whereas the opposite was seen in a study from a different group [57], where three out of four patients showed significant difference in VAS scores, compared to 3/6 MCS patients who experienced no pain relief. It is also possible that the relative merits of DBS and MCS may change depending on the specific aetiologies of the pain. For example, Katayama showed that for those with post-stroke pain, a patient may wish to opt for MCS over DBS; 48% (15/31) vs. 25% (3/10) success rate [56], but regarding phantom limb patients, DBS prevails with a success rate of 60% (6/10) in the DBS, but 20% (1/5) in MCS [55].

Safety comparisons between the two methods are much needed. The data is lacking, with the findings either unreported [55,56] or studies being simply too small to make a real comparison. To illustrate with Nandi et al., one DBS patient suffered a CSF leak and haematoma over the generator, whilst the MCS patient morbidity included one subdural haematoma and secondary wound infection, one seizure induced during post-op titration, one with a ‘strong motor response’ elicited during the procedure, and one patient ‘affected by the magnetic field’ with no further details given [57]. Whilst Nandi lists the adverse events, it is difficult to unpick a sensible conclusion, particularly if surgical skills and experience vary between the different modalities tested.

Explanations of why one patient is fielded into the MCS camp rather than DBS and vice-versa, are not provided. Furthermore, patient groups sometimes differ in baseline characteristics (Nandi et al. showed age characteristics of 59.5 years in the MCS group and 70.5 in the DBS group). Moreover, current publications are comparing MCS to a metaphorical and literal ‘moving target’ as electrode implants vary in locations. The targets of DBS may yield different results, such that sometimes the comparison is MCS vs. DBS:thalmus, other times MCS vs DBS: PAG/PVG and soon studies may compare MCS vs. DBS:ACC. This is particularly relevant if, as one meta-analysis demonstrates, the outcomes are more successful in those patients with targets in thalamus and PAG together, as compared to those patients with targets in the thalamus only; 87.3% success rate vs. 58%, respectively (*p* < 0.05) [61].

## 5. Conclusions

The use of neuromodulation for chronic pain has helped many patients, for whom pharmacotherapy has failed. SCS has been shown to be particularly useful for those with FBSS, but its success extends beyond this. DBS appears to fall short upon a review of the literature, possibly a misrepresentation of the innate potential of DBS as a treatment for chronic pain. The aforementioned methodological limitations of the published studies, together with the difficulty of comparing the efficacy to its closest alternatives, clouds the potential of DBS. On aggregate, it is difficult to draw conclusions from non-randomised trials that are understandably limited by small sample size. It may be that good-quality RCTs are not realistically achievable because of the cost of treatment and the rarity of its use. One alternative here is to use Bayesian statistics as has been suggested for rare disease groups. This approach provides probabilities of treatment effects of various percentages that can be applied to the next patient similar in clinical problems [137].

It is true that DBS does not reduce pain in all patients, and sometimes produces unwanted, mostly manageable, side effects. It is also true that many patients treated with DBS for chronic pain have been satisfied with their pain reduction, even some of those classed as a ‘failed treatment’ in the literature. Indeed, the National Institute of Clinical Excellence (NICE) guidelines do appreciate this, approving DBS for chronic refractory pain where other methods have failed and a multidisciplinary team of pain specialists approve of the case (IPG382). Although DBS for chronic pain is not currently funded on the National Health Service in the United Kingdom, it would seem that the world is experiencing a renaissance of the exploration of DBS for chronic pain; there are currently ten clinical trials registered regarding pain and DBS, with the United States claiming five of these, France four, and Denmark one suspended study (clinicaltrials.gov; nil found on EudraCT).

The results from previous papers and reviews suggest that the DBS for chronic pain is most successful for pain after amputation (both phantom limb and stump), FBBS, cranial and facial pain including anaesthesia dolorosa, and plexopathies. Poststroke pain is particularly successful if the type of pain reported is burning hyperaesthesia [138,139,140]. Given that we are dealing with refractory pain, rather than trying to ‘prove’ or ‘disprove’ the efficacy of DBS in large patient populations, perhaps it is more appropriate to adopt a treatment pathway that first uses less invasive therapies, followed by SCS or peripheral stimulation (if appropriate), and then DBS or MCS. Current pain theories would suggest over time, all of the pain circuits become centrally mediated, suggesting that a one-size-fits-all view may not be as inappropriate as previously thought, shifting the emphasis away from particular pain aetiologies being either amenable or unresponsive to DBS. It remains to be seen how the introduction of ACC as a target will change success rates for chronic pain patients, and how this will alter comparisons of DBS and MCS, as we await longer-term follow up data and increasing patient numbers. The targeting of an affective process promises a catch-all for those pain aetiologies that have proved more troublesome for less-invasive techniques. The obvious disadvantage being a possible risk of seizures/epilepsy.

The ability to pre-select individuals who respond well to a particular neuromodulation would lead to better outcomes. We are currently far from this patient-specific pre-selection ability, but some tantalising hints have proved simultaneously exciting and frustrating. Evidence from LFP recording shows chronic pain patients with DBS ‘off’ have characteristically enhanced low frequency (8–14 Hz) power spectra of both PAG and VP (thalamus) local field potentials when in pain [141]. Further research could explore non-invasive functional neuroimaging, including single-photo emission computed tomography, PET, and MEG to find correlates of this [109,142,143,144]. Perhaps rTMS may be an aid to selection as it can be with SCS and MCS. Meanwhile, there are suggestions that optimizing stimulation parameters post-surgery “through recursive testing and adjustments” leads to pain-improvement, with some evidence demonstrating optimal relief for two test patients with midbrain electrodes whilst cycling 2 Hz on for 1 s and off for 2 s, discovered during comprehensive meticulous parameter testing [145]. The possibilities for improving patient selection and success rates make this an exciting field of both research and clinical practice.

## Figures and Tables

**Table 1 brainsci-08-00158-t001:** Case studies and series of Deep Brain Stimulation targeting Anterior Cingulate Cortex.

Paper	Article Type	Patient N	Aetiology of Pain	Target	Outcome Measures	Follow up Times	Results	Conclusion
Boccard, Prangnell et al. (2017) [63]	case series	24	^a^FBSS (6), post-stroke (9), SCI (2), brachial plexus injury (3), unknown chest pain (1), head injury (1), ^b^RTA (2).	^c^ACC (bilateral)	^d^NRS, ^e^SF-36, EQ-^f^5D, ^g^MPQ	6 months, 1 year, 12 people f/u at 38.9, some at 42 months	At 6 months. NRS decreased from 8 to 4.27 (*p* = 0.004), MPQ improved (mean −36%; *p* = 0.021), EQ-5D score decreased (mean −21%; *p* = 0.036). The physical functioning domain of SF-36 was significantly improved (mean +54.2%; *p* = 0.01). At 1 year NRS score decreased by 43% (*p* < 0.01). EQ-5D reduced (mean −30.8; *p* = 0.01). Improvements in domains of SF-36. At the longer f/u; efficacy was sustained up to 42 months. NRS score as low as 3.	ACC stimulation alleviates chronic neuropathic pain refractory to pharmacotherapy.
Boccard, Fitzgerald et al. (2014) [124]	case series	16 (15 internalized; 11 followed up)	FBSS (6), Post-stroke (4), Spinal Cord Injury (1), Brachial plexus (3), unknown chest (1), head injury (1)	ACC (bilateral)	^h^VAS, SF-36, EQ-5D, McGill Pain Questionnaire	mean 13.2 months	Post-surgery, VAS decreased to <4 in five patients, and one patient reported to be pain free. Significant improvements on EQ-5D observed (mean 20.3%; range 0%–83%; *p* = 0.008). Statistically significant improvements were observed for the physical functioning and bodily pain domains of SF-36 quality of life survey; mean +64.7% (range, −8.9%–+27%; *p* = 0.015) and mean +39.0% (range −33.8%–+159%; *p* = 0.05).	ACC DBS can relieve chronic neuropathic pain refractory to pharmacotherapy and restore quality of life.
Boccard, Pereira et al. (2014) [123]	case study	1	RTA/brachial plexus injury	ACC (bilateral implants)	VAS, SF-36, McGill pain questionnaire, EQ-5D, Neuropsychological measures	2 years post-surgery	VAS decreased from 6.7 to 3; McGill improved by 43%, EQ-5D Health state increased by 150%.	ACC DBS efficacious; ACC target has potential for long-term control
Spooner, Yu et al. (2007) [122]	case report	1	Spinal Cord Injury at C4	ACC (bilateral); ^i^PVG (unilateral)	VAS, pain medication usage, described mood.	1–5 days post-surgery, 4 months post-surgery, 1 year not possible (patient died due to pulmonary issues)	Results most striking at 3 months with cingulum stimulus scoring VAS 3 and lidocaine usage of 2 (cc/hr), mood described as ‘best’. Compared to PVG (VAS 4, lidocaine 2, mood ‘average’) or no stimulation (VAS 10, lidocaine 5, mood ‘worst’).	Bilateral cingulate stimulation improved the patient’s mood and reduced pain more completely than PVG stimulation or medication alone

^a^FBSS = Failed Back Surgery Syndrome. ^b^RTA = Road Traffic Accident. ^c^ACC = Anterior Cingulate Cortex. ^d^NRS = Numeric Rating Scale. ^e^SF-36 = Short-form 36 quality of life questionnaire. ^f^EQ-5D = EuroQol 5-Domain quality of life questionnaire. ^g^MPQ = McGill Pain Questionnaire. ^h^VAS = Visual Analogue Scale. ^i^PVG = Periventricular Grey.

**Table 2 brainsci-08-00158-t002:** Studies involving comparison of Motor Cortex Stimulation and Deep Brain Stimulation.

Paper	Article Type	Patient N	Aetiology of Pain	Target	Outcome Measures	f/u Times	Results	Conclusion
Nandi et al. (2002) [57]	case-series	10	All post-stroke pain. ^a^MCS patients: post-stroke hemi-body pain (4); post-stroke facial pain (4); ^b^DBS patients: post-stroke hemi body (3), post-stroke face and leg (1)	^c^PVG	^d^VAS	2–3 weeks; some up to 4 years	MCS: 1/6 success rate. DBS:3/4 had at least 40% reduction in VAS scores during stimulation, 2/2 internalised with success.	MCS is not effective relieving post-stroke neuropathic pain. DBS is the preferred option.
Katayama et al (2001 a.) [55]	case-series	45	phantom limb (trauma- ^e^rt leg), brachial plexus avulsion (rt arm).	thalamus	VAS	unspecified- results reported to be ‘long term’	All 19 patients were given ^f^SCS and if failed were split into either DBS or MCS. For DBS 60% (6/10) gave pain relief, and for MCS 1/5 (20%) required pain relief. 4 patients were given both DBS and MCS- one patient reported better pain control by MCS than DBS. 2 patients reported the opposite.	DBS preferable to MCS, especially lower limb.
Katayama et al (2001 b.) [56]	case-series	45	post-stroke pain	thalamus	VAS	unspecified- results reported to be ‘long term’	Success rates (defined as >60% reduction in VAS scores) of 7% for SCS (3/45), 25% for DBS (3/12), 48% for MCS (15/31)	Success rate increases as stimulation moves higher. MCS more successful than DBS.
Son, Kim et al. (2014) [105]	open label	9^*^	Central post-stroke pain (4), ^g^SCI (4), amputation stump pain in arm (1)	ventralis caudalis (Vc) thalamus DBS	^h^NRS, medication use.	39 months mean, (8–72)	6/8 (75%) responded to MCS. 2/8 had successful DBS (one patient with amputation stump pain and the other with SCI pain caused by cervical syrinx). NRS score decreased significantly (*p* < 0.05) MCS: 37.9 ± 16.5 and DBS 37.5%.	Considering the initial success rate and the less invasive nature of epidural MCS compared with DBS, MCS would be a more reasonable initial means of treatment for chronic intractable neuropathic pain.

^a^MCS = Motor Cortex Stimulation. ^b^DBS = Deep Brain Stimulation. ^c^PVG = Periventricular Grey. ^d^VAS = Visual Analogue Scale. ^e^rt=right. ^f^SCS = Spinal Cord Stimulation. ^g^SCI = Spinal Cord Injury. ^h^NRS = Numeric Rating Scale. * = 8 successfully implanted and used in the comparison.

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
