# Peer review of "The Current State of Deep Brain Stimulation for Chronic Pain and Its Context in Other Forms of Neuromodulation"

_brainsci, 2018, doi:10.3390/brainsci8080158_

Round 1
Reviewer 1 Report
When DBS works in a chronic pain patient it is wonderful, but the high fraction of failures is very frustrating and has never been adequately explained. The failures may lead to abandonment of a critically important therapy with potential for a high success rate. This is an interesting and well argued paper, proposing that pain etiology may be less relevant to the success of DBS than other factors. Pre-selection based on better brain imaging (e.g DTI) and field potential measurements is mentioned at various points, along with psychological status (depression) and opioid tolerance as factors. That the risk:reward ratio for individuals is not derivable from mean success rates is another excellent point made. The importance of a multi-stage approach to therapy, employing increasing invasiveness, and of testing several brain sites per patient is also highlighted. I would suggest that the authors additionally consider flexibility post-internalization; by giving the patient long-term control of parameters, failures can flip to successes (Hentall et al., Brain Res. 1632:119-26).
Some specific issues that the authors may wish to address are as follows.
line 47. Shouldn't "upregulation" be "down-regulation"? Endogenous opioids are receptor agonists and the classic adaptive response to the long-term presence of agonists is down-regulation.
line 64. "This is neatly summarized by Chari et al.", etc., strikes a jarring note in an otherwise well written paper. What does "This" refer to? Why the naming and praising? I suggest a colon after "literature" in the prior sentence, followed by a simple listing starting with "the location". Also the word controversies may be less controversially expressed as differences or discrepancies or disagreements.
The abbreviation RCT should be stated one time in full.
Author Response
Thank you for reviewing this article. We are grateful for your constructive comments. We have endeavoured to address all as suggested.
- Flexibility post-internalisation is an excellent point and Hentall et al. has been added to the article as suggested.- lines 421-425
- “upregulation” has been changed to “downregulation”- line 47
- The unnecessary Chari et al. reference has been removed- line 64
- Addition of RCT stated in full- line 92
Many thanks for the comments, suggested additions and alterations.Reviewer 2 Report
This is an interesting review about the use of deep brain stimulation for the treatment of chronic pain.The authors summarize the existing literature of non-invasive and invasive central stimulation (spinal cord and brain). With regard of being a review paper with the topic of DBS and MCS relevant literature is not cited in my opinion. The authors should add papers of Rasche 2006; Levy and Adams 1987 to the DBS literature; Honey et al. 2017, Rasche 2006 and 2016 for MCS and Davis, KD J Neurosurg 2000 for the relationship of DBS and ACC; Finally diprenorphine-binding was already reported by Maarawi for MCS in 2007 and 2013.
Certainly the Targets addressing the affective component of chronic pain (N. Acc, ventral striatum or cingulum) are attractive new Stimulation sites. However the numbers of patients treated is small and the follow-up ist still quite short. Therefore a bit less optimistic view would be adequate.
Being a review some relevant papers a missing. Considering the Long-term use of DBS and MCS for more than 40 or 20 years respectively and the new Targets with 6 month to 2 year follow-up a more critical conclusion should be formulated. Otherwise verry nice paper!
Author Response
Thank you for reviewing this article. We are grateful for your constructive comments. We have endeavoured to address all as suggested.
The following suggested references have been added:
Rasche 2006, Levy and Adams 1987, Honey et al. 2017, Rasche 2006, Rasche 2016, Davis 2000, Maarrawi 2007 and Maarrawi 2013. These are all highlighted using the trackchanges comments box pointing to emboldened text in the new document.
The short f/u time and small number of patients with electrodes in newer areas, i.e affective processing, has been emphasized to provide less optimistic outlook. – line 295 and 41
Many thanks for the suggested additions and useful references.